# Parenting Styles Predict Future-Oriented Cognition in Children: A Cross-Sectional Study

**DOI:** 10.3390/children9101589

**Published:** 2022-10-20

**Authors:** Saeid Sadeghi, Sajad Ayoubi, Serge Brand

**Affiliations:** 1Institute for Cognitive and Brain Sciences, Shahid Beheshti University, Tehran 19839-69411, Iran; 2Psychiatric Clinics (UPK), Center of Affective, Stress and Sleep Disorders (ZASS), University of Basel, 4002 Basel, Switzerland; 3Faculty of Medicine, Department of Sport, Exercise and Health, Division of Sport Science and Psychosocial Health, University of Basel, 4002 Basel, Switzerland; 4Sleep Disorders Research Center, Kermanshah University of Medical Sciences, Kermanshah 67146, Iran; 5Substance Abuse Prevention Research Center, Kermanshah University of Medical Sciences, Kermanshah 67146, Iran; 6School of Medicine, Tehran University of Medical Sciences, Tehran 1416634793, Iran

**Keywords:** individual differences, parenting, cognitive development, future-oriented cognition

## Abstract

Parenting is a crucial environmental factor in children’s social and cognitive development. This study investigated the association between parenting styles and future-oriented cognition skills in elementary school-aged children. Cross-sectional data were collected from parents of 200 Iranian elementary school aged children (6–13 years), 139 boys and 61 girls. Baumrind’s Parenting Styles Questionnaire and Children’s Future Thinking Questionnaire (CFTQ) were administered to parents. There was a significant positive association between authoritative parenting and children’s abilities in prospective memory, episodic foresight, planning, delay of gratification, and future-oriented cognition total score. In contrast, authoritarian parenting was negatively correlated with children’s abilities in planning, delay of gratification, and future-oriented cognition. Increases in authoritative parenting scores predicted better future-oriented cognition abilities in children.

## 1. Introduction

Human beings can pre-experience and constantly anticipate future events [1]. Future-oriented cognition is one of the critical cognitive skills that children develop for daily performance that is built on memories, and it affects decision-making and mental health [2,3]. The term ‘future-orientation cognition’ has been used in a variety of ways, encompassing processes involved in constructing a coherent representation of the time, planning, goals, aspirations, hopes, worries, survival, and, ultimately, future prioritizing [2,3,4,5,6]. It has been shown that children can formulate imperfect plans at one year of age, and they begin to think about future events at three and four years of age [7,8]. Although children often have problems thinking accurately and also planning for the future by the age of five or six, future-oriented cognition improves in later years [2,7,9,10].

During children school years, orientation towards the future is vital in a range of daily functioning, including academic achievement (e.g., long-term planning for the final exam), social functioning (e.g., remembering the birthday of a close friend), and financial planning (e.g., saving money to meet future needs). Notably, negative and positive long-term outcomes associated with children’s future thinking extend far into their adulthood. For example, a research from Marroquín et al. [3] shows that pessimistic predictions are associated with psychopathology (e.g., suicide attempts), whereas better future-oriented skills (e.g., delay of gratification) are predictors of significant positive outcomes in mental and physical health [2,11].

There are many models that have attempted to explain the factors that predict cognitive development in children. Contrary to older models that highlighted the role of genetic factors in the development of cognitive skills [12], emerging ecological models suggest that cognitive abilities are developed within a combination of multilevel contextual and biological processes [13,14,15]. The underlying argument of the ecological models is that during critical periods, consistent environmental experiences impact the development of related circuity (prefrontal cortex) underpinning high-level cognitive skills [16]. According to this model, children are mainly dependent on their parents for providing opportunities to develop cognitive abilities through high-quality interactions [7,14,17,18,19]. One of the primary responsibilities of parents is raising their children [20]. As parents, we have different parenting styles that reflect our attitudes toward discipline, parenting responsibilities, and setting expectations of children [21].

In the first study of parenting styles [22], Diana Baumrind identified three groups of preschoolers with very different behaviors: (a) self-controlled, assertive, affiliative, self-reliant, and buoyant; (b) distrustful, withdrawn, and discontented; and (c) lack of self-reliance or self-control. Consequently, in the study of parental socialization [23], Baumrind developed a tripartite model of parenting styles: authoritative, authoritarian, and permissive. Indeed, these three styles corresponded to three modes of parental control, the authoritative control, the authoritarian control, and the lack of control (i.e., permissive control). Authoritative parents exert firm control over their children, expect maturity, and establish clear guidelines. Authoritarian parents emphasize conformity, compliance, parental control, respect for authority, and order. Additionally, permissive parents are known to be nurturing and clear in their communications but low in control [21]. Using the typologies of Baumrind, Eleanor Maccoby, and John Martin tested the generalizability of Baumrind typologies in families with a greater variety of socioeconomic backgrounds [24]. Instead of qualitatively separate categories, they conceptualized parental styles as measurable along two orthogonal dimensions: warmth and strictness [24,25,26,27,28,29,30,31,32,33]. Warmth indicates the degree to which the parents show their children love and affection; give them their support; communicate, talk, and reason with them [34,35,36]; and has a very similar meaning to acceptance/involvement, assurance, nurturance or love [29,30,37], responsiveness, involvement, acceptance or implication [26,34,38], or affection [34]. Strictness indicates the degree to which the parents use control and supervision, establish norms for children’s behavior, and maintain position of authority [26,39]. Other labels used in the literature are domination, hostility, inflexibility, firm control or restriction [29,33], demandingness, control, firmness [26,40], supervision [41], or imposition [42]. Four parental socialization styles result from the combination of these two main parental dimensions: authoritative (strictness and warmth); authoritarian (strictness but not warmth); indulgent (warmth but not strictness); and neglectful (neither strictness nor warmth) [24,26,31,34,37,42].

As defined, parenting styles are considered as essential elements in parenting research; they define how parents usually think, behave, and feel about raising their children [43,44,45]. In general, each of these parenting styles has a unique impact on children [21,46,47]. Social and cultural contexts often moderate parenting styles and all aspects of parenting are informed by culture [21]. Consequently, in the literature on parenting styles, there is a prominent controversy about which parenting style is associated with better child adjustment. Classical findings conducted in Anglo-Saxon contexts with European American samples (mostly white middle-class families) found benefits in authoritative parenting (i.e., warmth and strictness) [26,29,39,48]. However, it seems that the authoritative style is not always the best parenting. Other studies conducted in ethnic minorities in the United States, such as in Arab societies [49] and Chinese American societies (Chao, 2001), identify the benefits of the authoritarian parenting (i.e., strictness without warmth). The most recent studies on parental socialization, conducted in European and Latin American countries, support the idea that the optimal parenting style associated with better child adjustment is indulgent parenting [30,37,42,50,51,52,53].

Studies have shown that authoritative and positive parenting are significantly related to better cognitive and executive functions (EFs), for example, better attentional control, self-control, and better academic performance [14,54,55]. In a research from Gauvain and Huard [7], it has been suggested that authoritative parenting styles, parent–child interactions (the child is told to prepare game pieces before starting the game, verbal review of daily schedules), having more children in the family, and sibling interactions influence the planning ability of children. The study of Pinquart and Kauser [56] showed that children of authoritative parents have low levels of internalizing and externalizing behaviors. It was also shown by Pinquart [57] that authoritative parents can improve their children’s academic performance. However, Casas and Weigel [58] found that children raised by authoritarian parents often display hostility and shyness toward peers. Moreover, studies have shown that children of permissive parents often fail to control their impulses, lack self-reliance, and less academic success [29,59]. This is partly explained by a lack of self-reliance and decreased persistence on tasks [60]. In general, children with permissive parents are likely to be impulsive, demanding, selfish, and lacking in self-regulation [61,62].

Although early intensive environmental experiences, such as parenting, are associated with cognitive functions in an individual, we know little about the influences of parenting and environmental processes on children’s future-oriented cognition. To the best of our knowledge, the present study examines the relationship between parenting styles and children’s future-oriented cognition for the first time. Based on previous studies, we hypothesized that authoritative parents have futuristic children and parents with authoritarian parenting styles have less future-oriented cognition abilities. We also hypothesized that cognitive development depends on the opportunities provided by parents to their children. Finally, we explored how environmental factors (i.e., parenting and opportunities) influence children’s future-oriented cognition, as one of the most significant cognitive abilities.

## 2. Materials and Methods

### 2.1. Participants and Procedure

The study used a cross-sectional design. Participants were 200 Iranian parent–child pairs selected by convenience sampling. Parents were between the ages of 25 and 59 years old (mothers: *M* = 36.66, *SD* = 4.95: fathers: *M* = 40.98, *SD* = 5.26) and typical developing children (139 boys and 61 girls) aged between 6–13 years old (*M* = 9.06, *SD* = 1.79). The participants were from elementary schools (first grade: 57 students; second grade: 39 students; third grade: 36 students; fourth grade: 21 students; fifth grade: 22 students; and sixth grade: 25 students). Participants were studying in the academic year 2020–2021 and the online (Google Form) questionnaire along with the demographic checklist was provided to their parents. Inclusion criteria included study in primary school for children, no psychological and neurological disorders in the child (as reported by parents), and also parents’ familiarity with internet and having a smartphone or PC. Exclusion criteria included the failure to complete questioners and random answers to the questions.

### 2.2. Measures and Instruments

#### 2.2.1. Children’s Future Thinking Questionnaire

The Children’s Future Thinking Questionnaire (CFTQ) is considered as one of the most reliable and valid instruments for the assessment of future-oriented cognition development. Furthermore, the most recent Persian questionnaire, based on Mazachowsky and Mahy [2], was well validated and it was conducted in Iranian elementary school students [63]. Parents completed this questionnaire. The final form of the CFTQ had 44 items, which were aggregated into five dimensions: saving behavior (e.g., ‘saves a seat for someone who has not yet arrived (e.g., at the dinner table or at a play)’), prospective memory (e.g., ‘remembers what items need to be purchased/picked up (e.g., reminds parent to pick up cereal from grocery store)’), episodic foresight (e.g., ‘understands that he or she may be hungry later even though he or she has just eaten a large meal’), planning (e.g., ‘sets goals and takes steps to achieve those goals (e.g., wishes to learn to swim and asks parent to enroll him or her in swimming lessons)’), and delay of gratification (e.g., ‘forgoes a small treat in the present to receive a larger treat in the future (e.g., he or she would rather have two cookies after dinner vs. one cookie before dinner)’). Children’s capacity to save (e.g., money, objects, time, physical space) for future use was measured by the saving subscale. Children’s prospective memory subscale included items that assessed their abilities to remember to carry out their future intentions. A subscale measuring episodic foresight measured children’s ability to imagine, anticipate, and think about the future. In the planning subscale, children were measured on their ability to plan for the future and formulate goals. Finally, delay of gratification measured children’s ability to postpone gratification in the present in order to gain greater benefits in the future. This questionnaire assessed five dimensions on a 6-point Likert scale ranging from 1 (strongly disagree) to 6 (strongly agree). Cronbach’s alpha and Guttman’s split-half for the total score of the questionnaire were 0.89 and 0.85, respectively. The results of our recent study indicated the appropriate reliability of this questionnaire [63].

#### 2.2.2. Baumrind Parenting Style Inventory

The Baumrind Parenting Style Inventory was first developed by Burry in 1991 in order to assess parenting styles [64]. This inventory was 48-item, later it became 30-item; 10 items related to authoritative style (e.g., ‘I explain the reasons behind my expectations’), 10 items related to authoritarian style (e.g., ‘I use criticism to make my child improve his/her behavior’), and another 10 items related to permissive style (e.g., ‘I find it difficult to discipline my child’) [65]. There are five-point Likert-type ratings for each item (1 = Never, 5 = Always). A higher score indicates that behavior is more frequently used. The reliability of the three scales is good to excellent [66]. In a study by Minaei and Nikzad [67], the validity and reliability of 576 mothers of primary school students in Tehran has been reported with relatively good reliability. The current sample’s internal consistency is also good: Authoritative α = 0.83; Authoritarian α = 0.76; and Permissive α = 0.78.

### 2.3. Data Analysis

The data were summarized with descriptive statistics using SPSS_22_ for Windows. Correlations between variables were estimated by Pearson’s correlation. Additionally, associations between parenting styles and the children’s future-oriented cognition were estimated by linear regression.

## 3. Results

### 3.1. The Future-Oriented Cognition of Girls and Boys

Results showed that girls of primary school age have significantly higher planning (*t* = −3.24, *p* < 0.001), delay of gratification (*t* = −3.22, *p* < 0.001), and, in total, higher scores of future thinking ability (*t* = −2.68, *p* < 0.01) than boys (Table 1).

### 3.2. Correlation of Children’s Future-Oriented Cognition and Parenting Styles

Correlation coefficients between variables are provided in Table 2.

Children’s age was not correlated with their future-oriented cognition and their parent’s parenting style. The results showed that the authoritarian parenting was negatively correlated with children’s abilities for planning (*r*= −0.24, *p* ≤ 0.01), delay of gratification (*r*= −0.15, *p* ≤ 0.05), and total score of future-oriented cognition (*r*= −0.23, *p* ≤ 0.01). In contrast, authoritative parenting was positively correlated with children’s abilities for prospective memory (*r*= 0.29, *p* ≤ 0.01), episodic foresight (*r*= 0.19, *p* ≤ 0.01), planning (*r*= 0.30, *p* ≤ 0.01), delay of gratification (*r*= 0.14, *p* ≤ 0.05), and also the total score of future-oriented cognition (*r*= 0.27, *p* ≤ 0.01). See Figure 1, Figure 2 and Figure 3 for the relationship between parenting styles and total future-oriented cognition scores in children.

### 3.3. A regression Modeling of Relationship between Children’s Future-Oriented Cognition and Parenting Styles

Linear regression modeling was used to make predictions of future-oriented cognition (Table 3).

For each unit increase in the authoritative parenting score, the future-oriented cognition total score increased by 0.08 (*F* = 10.97, *p* < 0.001). Likewise, for each unit increase in the authoritative parenting score, the prospective memory score increased by 0.09 (*F*= 18.71, *p* < 0.001), episodic foresight score increased by 0.04 (*F*= 7.48, *p* < 0.01), planning score increased by 0.09 (*F* = 20.04, *p* < 0.0001), and delay of gratification increased by 0.02 (*F* = 4.15, *p* < 0.05). No significant association between authoritative parenting score and saving scores (*F*= 2.66, *p* > 0.05) was found. Additionally, results indicated an increase in authoritarian parenting score for each unit, whereas future-oriented cognition total score and planning score decreased by 0.05 (*F* = 7.80, *p* < 0.01) and 0.06 (*F*= 12.28, *p* < 0.01), respectively.

## 4. Discussion

The study aimed to investigate parenting styles correlation with children’s future-oriented cognition. This study highlighted new directions for research into parenting and children’s cognitive outcomes. The findings of the study confirmed our hypothesis. The results of this study showed that parenting styles (authoritative, authoritarian, and permissive) are good predictors of children’s future-oriented cognitions, such as saving, prospective memory, episodic foresight, planning, and delay of gratification. Consistent with these findings, a study by Bindman and Pomerantz [68] demonstrated that parenting influences executive functions (i.e., delay of gratification). Accordingly, family environment and parenting behaviors influence children’s executive functions [14,15]. The findings of this study are also congruent with those of a study by Töz and Arikan [69], which showed a positive correlation between specific parenting style (authoritative), socioeconomic status (SES), and parents’ education and cognitive abilities development (i.e., general cognitive developments and prospective memory). This study also revealed a negative correlation between authoritarian parenting style, SES, and parents’ educational level and cognitive abilities (i.e., planning, delay of gratification). We demonstrated that elementary school-aged children’s age was not correlated with their future-oriented cognition (i.e., saving, prospective memory, episodic foresight, planning, and delay of gratification), while previous studies showed an age-related increase in saving behavior [70], prospective memory [71], episodic foresight [72], planning [73], and delay of gratification [74] of preschool children. This inconsistency between our results and previous studies can be explained by two reasons: First, our sample group consisted of elementary school-age children, who can be expected to differ from preschool children in terms of cognitive developmental pathways. Second, this observed difference in results could be due to differences in the measures used. Most of the previous studies of preschool children discussed above used different tasks than our study. Experimental studies and cross-sectional studies with a larger sample size are needed to study the evolution of future-oriented cognition in elementary-school children.

Additionally, we found gender differences in future-oriented cognition: girls in primary school age have significantly higher planning, delay of gratification ability, and better total score in future thinking ability than boys. Although the sample in this study was not gender balanced, this finding is notable. In contrast to these results, the study by Mazachowsky and Mahy [2] found no gender differences in the future-oriented cognition of preschool children. This inconsistency in results may be due to the difference in the age group of our study and the study by Mazachowsky and Mahy [2]. Our result is consistent with the study by Barnett and Heron [75] of in a normative sample of 8- and 10-year-old children, who reported that girls had better attention scores than boys, tested with a standard neuropsychological battery. Differences in structural brain development may contribute to explaining gender differences in future-oriented cognition as well as executive functioning. The prefrontal cortex (PFC) has received a lot of attention in terms of executive function, with links to attention, impulsivity, and working memory, including both the medial PFC (mPFC) [76,77] and the orbitofrontal cortex (oPFC) [78,79]. There is evidence that the development of the prefrontal cortex occurs later in males than in females, although those of males are consistently greater in volume and thickness [80,81], particularly through adolescence. The trajectories of functional connectivity between the left and right dorsolateral prefrontal cortices also differ between females and males, with a pattern suggesting earlier maturation in females [82]. Thus, the difference between future-oriented cognition in male and female subjects in this study can be explained by the gender differences in the developmental trajectory of the prefrontal cortex.

According to the framework of parenting and environmental factors, the vast majority of research found authoritative parenting style as a consistent predictor of positive outcomes in children and adolescences [5,7,43,64,67,83,84]. These authors reported that authoritative parenting styles impact children’s socio/cognitive abilities. In these studies, parenting style has an important role in a child joining in planning-related discussions, social responsibilities, preparing for self-regulation behaviors, emotion-regulations, creativity, academic achievement, clear moral boundaries, and also better future-oriented behaviors. To strengthen parenting style research findings, future research needs to consider contextual variability. For instance, association between authoritative parenting style and school performance is stronger among European American and Hispanic American students than Asian American and African American students [26].

In summary, our study’s findings are consistent with classical studies conducted in Anglo-Saxon contexts with European American samples (mostly white middle-class families). According to these classical investigations, authoritative parenting is the ideal parenting style [29,39,48]. However, more recent studies on European American samples show different findings. For example, most recent literature on parenting in European and Latin American countries confirms that the optimal parenting style associated with better child adjustment is indulgent parenting (i.e., higher warmth but low strictness) [30,37,42,50,51,52,53]. It seems that parenting patterns and their psychological consequences differ from one culture to another and over time. The results of the present study provided evidence about how the parenting styles and environmental factors affect cognitive and behavioral outcomes. It is better to consider other variables as well, because in addition to parenting, other variables also have the potential to influence children’s futurism. However, it should be noted that child behavior can also determine parental behavior, and it is better to pay attention to this bidirectional relationship—thus longitudinal studies and other functional measurement tools are recommended in future studies. Additionally, the sample in this study is not gender balanced. Future studies should have a balanced gender sample in order to make comparisons.

## Figures and Tables

**Figure 1 children-09-01589-f001:**
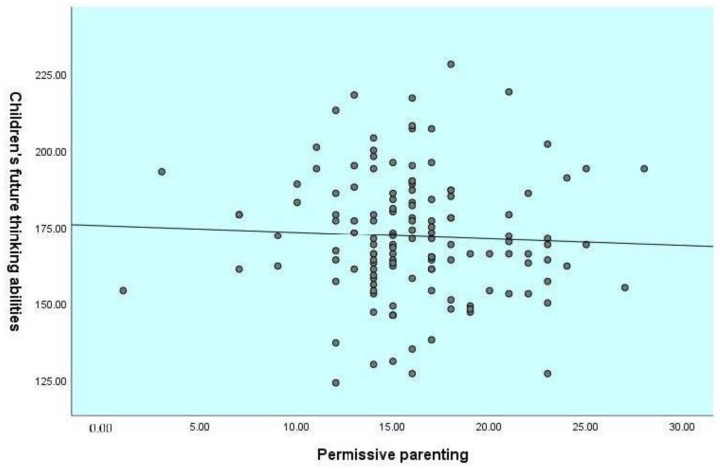
Correlation between permissive parenting and children’s future-oriented cognition total score.

**Figure 2 children-09-01589-f002:**
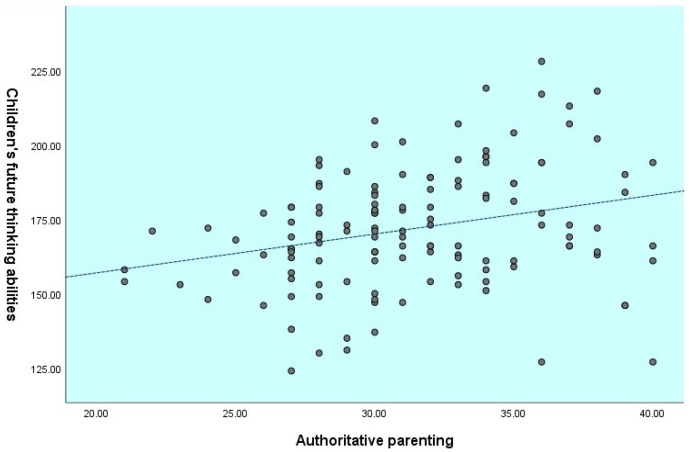
Correlation between authoritative parenting and children’s future-oriented cognition total score.

**Figure 3 children-09-01589-f003:**
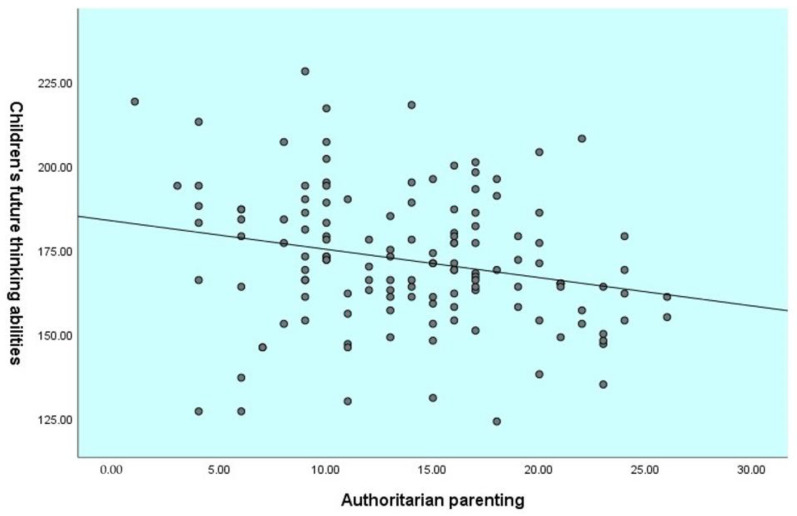
Correlation between authoritarian parenting and children’s future-oriented cognition total score.

**Table 1 children-09-01589-t001:** Descriptive statistics of the study measures and a comparison between males’ and females’ future-oriented cognition skills and their parents’ parenting style (*n* = 200).

Variable		Descriptive Statistics				Compare Groups	
		*M*		*SD*		*t*	*p*
		Male	Female	Male	Female		
Parenting style	Permissive	15.56	14.72	4.64	3.96	1.23	0.22
	Authoritarian	13.92	12.47	6.09	5.39	1.60	0.11
	Authoritative	31.62	32.36	4.64	3.70	−1.10	0.27
Future-oriented cognition	Saving	33.23	33.85	4.92	4.92	−0.68	0.50
	Prospective Memory	35.89	37.06	5.92	4.54	−1.38	0.17
	Episodic Foresight	35.32	37.08	6.66	6.71	−1.72	0.09
	Planning	37.99	41.52	7.28	6.63	−3.24	0.001
	Delay of Gratification	31.43	34.69	6.59	6.54	−3.22	0.001
	Total score	168.85	178.68	19.10	20.69	−2.68	0.008

Abbreviations: *M*, Mean: *SD*, Standard deviation: *t*, independent *t*-test: *p*, Significance level.

**Table 2 children-09-01589-t002:** Correlation coefficients between variables.

Variables		1	2	3	4	5	6	7	8	9	10
	Age	1									
**Parenting**	2. Permissive	0.01	1								
	3. Authoritarian	0.09	0.07	1							
	4. Authoritative	−0.07	0	−0.44 **	1						
**Future Thinking**	5. Saving	−0.05	−0.04	−0.12	0.14	1					
	6. Prospective Memory	0.07	−0.12	−0.14	0.29 **	0.17 *	1				
	7. Episodic Foresight	0.06	−0.12	−0.06	0.19 **	0.14	0.35 **	1			
	8. Planning	0.06	−0.14	−0.24 **	0.30 **	0.28 **	0.54 **	0.66 **	1		
	9. Delay of Gratification	0.09	−0.07	−0.15 *	0.14 *	0.27 **	0.20 **	0.46 **	0.48 **	1	
	10. Total score	−0.01	−0.05	−0.23 **	0.27 *	0.51 **	0.60 **	0.73 **	0.84 **	0.66 **	1

Abbreviations: *, *p* ≤ 0.05: **, *p* ≤ 0.01.

**Table 3 children-09-01589-t003:** Linear regression for parenting style predicting future-oriented cognition in children.

Predicted Variable: Future-Oriented Cognition	Predictor Variables: Parenting Style								
	Permissive Parenting			Authoritarian Parenting			Authoritative Parenting		
	*R* ^2^	*F*	B (SE)	*R* ^2^	*F*	B (SE)	*R* ^2^	*F*	B (SE)
Saving	0.002	0.27	−0.45 (0.10)	0.01	1.88	−0.10 (0.07)	0.01	2.66	0.16 (0.10)
Prospective Memory	0.01	2.76	−0.12 (0.09)	0.02	3.73	−0.13 (0.07)	0.09	18.71 ***	0.37 (0.09)
Episodic Foresight	0.01	2.77	−0.12 (0.11)	0	0.68	−0.07 (0.08)	0.04	7.48 **	0.29 (0.11)
Planning	0.02	4.25	−0.14 (0.11)	0.06	12.28 **	−0.30 (0.08)	0.09	20.04 ***	050 (0.11)
Delay of Gratification	0	1	−0.07 (0.11)	0.02	4.40	−0.17 (0.08)	0.02	4.15 *	0.22 (0.11)
Total score	0	0.30	−0.05 (0.40)	0.05	7.80 **	−0.84 (0.30)	0.08	10.97 ***	1.30 (0.39)

* *p* ≤ 0.05, ** *p* < 0.01, *** *p* ≤ 0.01.

## Data Availability

The data that support the findings of this study are available on request from the corresponding author.

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
