# Peer review of "Parenting Styles Predict Future-Oriented Cognition in Children: A Cross-Sectional Study"

_children, 2022, doi:10.3390/children9101589_

Round 1

Reviewer 1 Report

Thank you for inviting me to review this paper, which is an innovative and interesting study. However, I have some suggestions for the authors that might help strengthen their manuscript.

Introduction:

The article would be improved if it presented a more in-depth theoretical framework and was supported by more empirical evidence. For example, it is advisable to extend the framework on parental styles, and especially to better describe the two orthogonal dimensions of parental behavior, demandingness and responsiveness, as well as theoretical developments subsequent to Baumrind's on parental styles (e.g. Maccoby & Martin, 1983; Steinberg, 2005…)

Materials and Methods

·        This section could be improved by structuring it as follows:

Materials and Methods

2.1. Participants and Procedure

2.2. Measures and Instruments

2.3. Data Analysis

·        The description of the sample in this section should be further clarified. Important data such as gender and age distribution are missing.

·        Could you justify why the sample is not gender balanced, with more than twice as many boys as girls?

·        Could you explain why "“no severe sensory problems in the child and parent (blindness and deafness)” was taken as an inclusion criteria?

·        It is important to provide a more exhaustive description of the measurement instruments, for instance, examples of some items of each dimension of the questionnaires should be presented, as well as the psychometric properties of the Baumrind Parental Styles Inventory.

Results

·           It is recommended to organize the results in subheadings, according to the variables analyzed.

·        It is necessary to provide more information to better develop this section, interspersing numerical data with the presentation of the results in text (e.g., as is done with the results in Table 5).

     Table 1 could be deleted, explaining the basic data of the sample description within the text of section 2 (materials and methods), and describing, as noted above, not only the age distribution of the sample, but also the sex distribution.

     Table 2 can also be left out if it does not provide relevant information for the study, simplifying the description of the sample also in section 2. It is advisable to take into account the relevance of these data when deciding whether or not to display them graphically.

Discussion

If possible, the results should be developed, providing further empirical evidence congruent with the results, and contrasting them with other research that finds different results.

Author Response

We thank Reviewer #1 for their valuable comments, which helped us to improve the quality of the manuscript. Please find attached the detailed point-by-point-response. Once again, thank you so much for the care devoted to the present manuscript. 

Reviewer 2 Report

Manuscript Number children-1912529

Title: “Parenting styles predict future-orientation cognition in children: A cross-sectional study”

Present study aims to investigate the association between parenting styles and future-oriented cognition skills in elementary school-aged children. Participants were parents of 200 Iranian elementary school aged (6 – 13 years) and the children were studying in the academic year 2020-2021. Parents responded the online (Google Form) questionnaire along with the demographic checklist, the Children's Future Thinking Questionnaire (CFTQ) and the Baumrind Parenting Style Inventory. The analysis plan consisted of correlation analysis and linear regression analysis. Correlation analysis was performed between parenting styles and future-oriented cognition in children variables. Linear regression analysis was performed with parenting styles as predictor variables and future-oriented cognition in children variables as predicted variables. Results from correlation analysis and linear regression analysis were congruent with each other. There was a significant positive association between authoritative parenting and future-oriented cognition skills in children. A negative relationship between authoritarian parenting and future-oriented cognition skills in children was also found.

The focus of the paper is quite interesting. However, there are important theoretical and empirical weakness that should be addressed within the text of the manuscript to improve its quality. The introduction is quite poor, and it is not adequately rationalized. This research examines parenting without discussing the extensive classic and recent literature on the subject.

Theoretical part

Further theoretical conceptualization of the classic and recent literature on parenting styles should be added in the Introduction section. Authors should discuss the three parenting styles of Baumrind's tripartite model and, the theoretical framework proposed by Maccoby and Martin (1983), who became the referential model for the study of parental socialization.

On the one hand, the manuscript describes the three parenting styles proposed by Baumrind based on the two parental dimensions proposed by Maccoby and Martin. It should be noted that Maccoby and Martin's model is later than Baumrind's model. Therefore, the authors should conceptualize Baumrind's model as Baumrind did in her studies.

Parents have, as one of the main responsibilities, raising children (Sandoval-Obando et al., 2022). In this regard, in the study of parental socialization, Diana Baumrind defined a tripartite model yielding three parenting styles: authoritative, authoritarian, and permissive; these three parenting patterns corresponded to three modes of parental control, the authoritative control, the authoritarian control, and the lack of control (i.e., permissive control) (Baumrind, 1968). Recently, the permissive style of Baumrind's tripartite model has been called the indulgent style.

On the other hand, the bidimensional model proposed by Maccoby and Martin (Maccoby & Martin, 1983) became the referential model for the study of parental socialization.  The two main theoretically orthogonal dimensions stated by Maccoby and Martin are warmth and strictness (Baumrind, 1991a; Darling & Steinberg, 1993; F. Garcia & Gracia, 2009; Gimenez-Serrano et al., 2021; Lamborn et al., 1991; Maccoby & Martin, 1983; I. Martinez et al., 2019; I. Martinez et al., 2020; Martinez-Escudero et al., 2020; Martínez et al., 2012; Martínez et al., 2021).

Warmth indicates the degree to which the parents show the children love and affection, give them their support, communicate, talk and reason with them (Axpe et al., 2019; I. Martinez et al., 2020; Martínez et al., 2019) and has very similar meaning to acceptance/involvement, assurance, nurturance or love (Gimenez-Serrano et al., 2021; Lamborn et al., 1991; I. Martinez et al., 2019) responsiveness, involvement, acceptance or implication (Darling & Steinberg, 1993; F. Garcia & Gracia, 2014; I. Martinez et al., 2020), or affection (Martinez-Escudero et al., 2020). 

Strictness indicates the degree to which the parents use control and supervision, establish norms for children’s behavior, and maintain position of authority (Baumrind, 1991c; Darling & Steinberg, 1993). Other labels used in the literature are domination, hostility, inflexibility, firm control or restriction (Lamborn et al., 1991; Martínez et al., 2017), demandingness, control, firmness (Darling & Steinberg, 1993; Steinberg, 2005), supervision (O. F. Garcia et al., 2020), or imposition (Martinez-Escudero et al., 2020).

From the combination of the two main parental dimensions emerge the four parental socialization styles: authoritative (strictness and warmth); authoritarian (strictness but not warmth); indulgent (warmth but not strictness); and neglectful (neither strictness nor warmth) (Darling & Steinberg, 1993; O. F. Garcia et al., 2020; Gimenez-Serrano et al., 2022; Maccoby & Martin, 1983; Martinez-Escudero et al., 2020; Martínez et al., 2021).

Maccoby and Martin's (1983) two-dimensional model has had wide repercussions on the literature about parental socialization and has given rise to many studies of child psychosocial adjustment as a function of the two main parenting dimensions (O. F. Garcia et al., 2020; Gimenez-Serrano et al., 2022; Martínez et al., 2021; Perez-Gramaje et al., 2020).

It is very important to discuss a prominent controversy within the literature on parental socialization. It is about which parenting style is associated with better child adjustment.

Classical findings conducted in Anglo-Saxon contexts with European-American samples (mostly white middle-class families) found the benefits of the authoritative parenting (i.e., warmth and strictness) (Baumrind, 1991b; Lamborn et al., 1991; Steinberg et al., 1994).

However, it seems that the authoritative style is not always the best parenting. Other studies conducted in ethnic minorities in the United States such as in Arabs societies (Dwairy & Achoui, 2006) and Chinese-American societies (Chao, 2001) identify the benefits of the authoritarian parenting (i.e., strictness without warmth).

The most recent studies on parental socialization, conducted in European and Latin American countries, support the idea that the optimal parenting style associated with better child adjustment is the indulgent parenting (i.e., higher warmth but low strictness) (Fuentes et al., 2020; O. F. Garcia & Serra, 2019; Gimenez-Serrano et al., 2022; Martinez-Escudero et al., 2020; Martínez et al., 2019; Perez-Gramaje et al., 2020; Villarejo et al., 2020).

In Discussion section, authors should add literature agree and disagree to the present findings. Present findings are agreed to classical studies conducted in Anglo-Saxon contexts with European-American samples (mostly white middle-class families). These investigations affirm that the optimal parenting style is authoritative (Baumrind, 1991b; Lamborn et al., 1991; Steinberg et al., 1994). The present results do not coincide with the most recent literature on parenting in European and Latin American countries, which affirms that the optimal parenting style associated with better child adjustment is indulgent parenting. (i.e., higher warmth but low strictness) (Fuentes et al., 2020; O. F. Garcia & Serra, 2019; Gimenez-Serrano et al., 2022; Martinez-Escudero et al., 2020; Martínez et al., 2019; Perez-Gramaje et al., 2020; Villarejo et al., 2020).

Empirical part

First, authors should add more details about the measures.

On the one hand, the authors should provide more details about the application process of the Children's Future Thinking Questionnaire (CFTQ). Is it indicated for the age range determined in elementary school-aged children and therefore answered by parents? Is it applicable to any age and self-reported? In this research has it been decided that parents answer about their children in a hetero-informed measure?

In the same vein, what is meant by "Parents completed this questionnaire as a self-report" in the text? This does not seem to be very clear since a self-report measure implies that the respondent reports about him/herself and in the present research the adjustment of the child is studied but not that of the parents.

Besides, the authors should add an example of an item from each of the dimensions assessed by the Children's Future Thinking Questionnaire (CFTQ) to facilitate understanding of the content assessed.

On the other hand, in the Baumrind Parenting Style Inventory, the authors should add an example item for each of the parenting styles assessed to facilitate understanding of the content assessed.

Second, some changes need to be made in the results section.

Statistics should appear in italics, for example, where (M = 39 SD = 5.4) appears, it should appear (M = 39 and SD = 5.4).

As the same results have been obtained with correlation and regression analysis, it could be specified in the results.

In the results section, in linear regression, the authors use the labels dependent and independent variables. Instead of using these labels, the authors could refer to these concepts as predicted variables and predictor variables, respectively.

References

Axpe, I., Rodriguez-Fernandez, A., Goni, E., & Antonio-Agirre, I. (2019). Parental socialization styles: The contribution of paternal and maternal affect/communication and strictness to family socialization style. International Journal of Environmental Research and Public Health, 16(12), 2204. https://doi.org/10.3390/ijerph16122204

Baumrind, D. (1968, Authoritarian vs authoritative parental control. Adolescence, 3, 255-272.

Baumrind, D. (1991a). Effective parenting during the early adolescent transition. In P. A. Cowan, & E. M. Herington (Eds.), Advances in family research series. Family transitions (pp. 111-163). Lawrence Erlbaum Associates, Inc.

Baumrind, D. (1991b). The influence of parenting style on adolescent competence and substance use. Journal of Early Adolescence, 11(1), 56-95. https://doi.org/10.1177/0272431691111004

Baumrind, D. (1991c). Parenting styles and adolescent development. In R. M. Lerner, A. C. Petersen & J. Brooks-Gunn (Eds.), Encyclopedia of adolescence (pp. 746-758). Garland.

Chao, R. K. (2001). Extending research on the consequences of parenting style for Chinese Americans and European Americans. Child Development, 72, 1832-1843. https://doi.org/10.1111/1467-8624.00381

Darling, N., & Steinberg, L. (1993). Parenting style as context: An integrative model. Psychological Bulletin, 113(3), 487-496. https://doi.org/10.1037/0033-2909.113.3.487

Dwairy, M., & Achoui, M. (2006). Introduction to three cross-regional research studies on parenting styles, individuation, and mental health in Arab societies. Journal of Cross-Cultural Psychology, 37, 221-229. https://doi.org/10.1177/0022022106286921

Fuentes, M. C., Garcia, O. F., & Garcia, F. (2020). Protective and risk factors for adolescent substance use in Spain: Self-esteem and other indicators of personal well-being and ill-being. Sustainability, 12(15), 5967. https://doi.org/10.3390/su12155962

Garcia, F., & Gracia, E. (2014). The indulgent parenting style and developmental outcomes in South European and Latin American countries. In H. Selin (Ed.), Parenting Across Cultures (pp. 419-433). Springer. https://doi.org/10.1007/978-94-007-7503-9_31

Garcia, F., & Gracia, E. (2009). Is always authoritative the optimum parenting style? Evidence from Spanish families. Adolescence, 44(173), 101-131.

Garcia, O. F., Fuentes, M. C., Gracia, E., Serra, E., & Garcia, F. (2020). Parenting warmth and strictness across three generations: Parenting styles and psychosocial adjustment. International Journal of Environmental Research and Public Health, 17(20), 7487. https://doi.org/10.3390/ijerph17207487

Garcia, O. F., & Serra, E. (2019). Raising children with poor school performance: Parenting styles and short- and long-term consequences for adolescent and adult development. International Journal of Environmental Research and Public Health, 16(7), 1089. https://doi.org/10.3390/ijerph16071089

Gimenez-Serrano, S., Alcaide, M., Reyes, M., Zacarés, J. J., & Celdrán, M. (2022). Beyond parenting socialization years: The relationship between parenting dimensions and grandparenting functioning. International Journal of Environmental Research and Public Health, 19(8) https://doi.org/10.3390/ijerph19084528

Gimenez-Serrano, S., Garcia, F., & Garcia, O. F. (2021). Parenting styles and its relations with personal and social adjustment beyond adolescence: Is the current evidence enough? European Journal of Developmental Psychology, 19(5), 749-769. https://doi.org/10.1080/17405629.2021.1952863

Lamborn, S. D., Mounts, N. S., Steinberg, L., & Dornbusch, S. M. (1991). Patterns of competence and adjustment among adolescents from authoritative, authoritarian, indulgent, and neglectful families. Child Development, 62(5), 1049-1065. https://doi.org/10.1111/j.1467-8624.1991.tb01588.x

Maccoby, E. E., & Martin, J. A. (1983). Socialization in the context of the family: Parent–child interaction. In P. H. Mussen (Ed.), Handbook of child psychology (pp. 1-101). Wiley.

Martínez, I., Cruise, E., Garcia, O. F., & Murgui, S. (2017). English validation of the Parental Socialization Scale—ESPA29. Frontiers in Psychology, 8(865), 1-10. https://doi.org/10.3389/fpsyg.2017.00865

Martínez, I., Garcia, F., Musitu, G., & Yubero, S. (2012). Family socialization practices: Factor confirmation of the Portuguese version of a scale for their measurement. Revista De Psicodidactica, 17(1), 159-178. https://doi.org/10.1387/RevPsicodidact.1306

Martínez, I., Murgui, S., Garcia, O. F., & Garcia, F. (2021). Parenting and adolescent adjustment: The mediational role of family self-esteem. Journal of Child and Family Studies, 30(5), 1184-1197. https://doi.org/10.1007/s10826-021-01937-z

Martinez, I., Garcia, F., Fuentes, M. C., Veiga, F., Garcia, O. F., Rodrigues, Y., Cruise, E., & Serra, E. (2019). Researching parental socialization styles across three cultural contexts: Scale ESPA29 bi-dimensional validity in Spain, Portugal, and Brazil. International Journal of Environmental Research and Public Health, 16(2), 197. https://doi.org/10.3390/ijerph16020197

Martínez, I., Murgui, S., Garcia, O. F., & Garcia, F. (2019). Parenting in the digital era: Protective and risk parenting styles for traditional bullying and cyberbullying victimization. Computers in Human Behavior, 90, 84-92. https://doi.org/10.1016/j.chb.2018.08.036

Martinez, I., Garcia, F., Veiga, F., Garcia, O. F., Rodrigues, Y., & Serra, E. (2020). Parenting styles, internalization of values and self-esteem: A cross-cultural study in Spain, Portugal and Brazil. International Journal of Environmental Research and Public Health, 17(7), 2370. https://doi.org/10.3390/ijerph17072370

Martinez-Escudero, J. A., Villarejo, S., Garcia, O. F., & Garcia, F. (2020). Parental socialization and its impact across the lifespan. Behavioral Sciences, 10(6), 101. https://doi.org/10.3390/bs10060101

Perez-Gramaje, A. F., Garcia, O. F., Reyes, M., Serra, E., & Garcia, F. (2020). Parenting styles and aggressive adolescents: Relationships with self-esteem and personal maladjustment. European Journal of Psychology Applied to Legal Context, 12(1), 1-10. https://doi.org/10.5093/ejpalc2020a1

Sandoval-Obando, E., Alcaide, M., Salazar-Muñoz, M., Peña-Troncoso, S., Hernández-Mosqueira, C., & Gimenez-Serrano, S. (2022). Raising children in risk neighborhoods from Chile: Examining the relationship between parenting stress and parental adjustmenthttps://doi.org/10.3390/ijerph19010045

Steinberg, L. (2005). Psychological control: Style or substance? In J. G. Smetana (Ed.), New directions for child and adolescent development: Changes in parental authority during adolescence (pp. 71-78). Jossey-Bass. https://doi.org/10.1002/cd.129

Steinberg, L., Lamborn, S. D., Darling, N., Mounts, N. S., & Dornbusch, S. M. (1994). Over-Time changes in adjustment and competence among adolescents from authoritative, authoritarian, indulgent, and neglectful families. Child Development, 65(3), 754-770. https://doi.org/10.1111/j.1467-8624.1994.tb00781.x

Villarejo, S., Martinez-Escudero, J. A., & Garcia, O. F. (2020). Parenting styles and their contribution to children personal and social adjustment. Ansiedad y Estrés, 26(1), 1-8. https://doi.org/10.1016/j.anyes.2019.12.001

Author Response

We thank Reviewer #2 for their valuable comments, which helped us to improve the quality of the manuscript. Please find attached the detailed point-by-point-response. Once again, thank you so much for the care devoted to the present manuscript. 

Round 2

Reviewer 2 Report

For my part the review is satisfactory. The authors have responded to all my concerns.

Author Response

Once again, thank you so much for the care devoted to the revised version of the manuscript. 
